# Spasticity Management in Disorders of Consciousness

**DOI:** 10.3390/brainsci7120162

**Published:** 2017-12-09

**Authors:** Géraldine Martens, Steven Laureys, Aurore Thibaut

**Affiliations:** Coma Science Group, GIGA Research (Interdisciplinary Cluster for Applied Genoproteomics)–GIGA Consciousness & Neurology Department, University and University Hospital of Liege, 4000 Liege, Belgium; steven.laureys@ulg.ac.be (S.L.); athibaut@ulg.ac.be (A.T.)

**Keywords:** spasticity, pain, upper motor neuron syndrome, disorders of consciousness, brain injury, treatment

## Abstract

**Background:** Spasticity is a motor disorder frequently encountered after a lesion involving the central nervous system. It is hypothesized to arise from an anarchic reorganization of the pyramidal and parapyramidal fibers and leads to hypertonia and hyperreflexia of the affected muscular groups. While this symptom and its management is well-known in patients suffering from stroke, multiple sclerosis or spinal cord lesion, little is known regarding its appropriate management in patients presenting disorders of consciousness after brain damage. **Objectives:** Our aim was to review the occurrence of spasticity in patients with disorders of consciousness and the therapeutic interventions used to treat it. **Methods:** We conducted a systematic review using the PubMed online database. It returned 157 articles. After applying our inclusion criteria (i.e., studies about patients in coma, unresponsive wakefulness syndrome or minimally conscious state, with spasticity objectively reported as a primary or secondary outcome), 18 studies were fully reviewed. **Results:** The prevalence of spasticity in patients with disorders of consciousness ranged from 59% to 89%. Current treatment options include intrathecal baclofen and soft splints. Several treatment options still need further investigation; including acupuncture, botulin toxin or cortical activation by thalamic stimulation. **Conclusion:** The small number of articles available in the current literature highlights that spasticity is poorly studied in patients with disorders of consciousness although it is one of the most common motor disorders. While treatments such as intrathecal baclofen and soft splints seem effective, large randomized controlled trials have to be done and new therapeutic options should be explored.

## 1. Introduction

Following a severe brain injury, patients may suffer from disorders of consciousness (DOC), encompassing unresponsive wakefulness syndrome/vegetative state (UWS/VS)—meaning the patient shows awareness without any consciousness of self or the environment [1], minimally conscious state (MCS)—meaning the patient shows fluctuating behavioral signs of consciousness such as response to command or visual pursuit, or emergence from the minimally conscious state (EMCS)—meaning the patient is able to functionally communicate or use objects appropriately [2]. They often face a significant amount of functional, cognitive and motor impairment. By definition, these patients are unable to communicate, and therefore cannot express if they are uncomfortable or if they are suffering. This is why it is essential to detect and treat potential sources of pain, such as spasticity. Up to 89% of DOC patients suffer from spasticity, a motor disorder occurring following a lesion of the pyramidal tract [3]. This condition is usually defined as “a motor disorder, characterized by a velocity-dependent increase in tonic stretch reflexes (muscle tone) with exaggerated tendon jerks, resulting from hyper-excitability of the stretch reflex as one component of the upper motor neuron (UMN) syndrome” [4]. Other associated symptoms are increased hypertonia, altered sensori-motor control and muscle spasms, scissoring (involuntary crossing of the legs) and fixed joints [5,6]. Several treatment options have already been investigated for various conditions (e.g., stroke, spinal cord injury or multiple sclerosis) with favorable outcomes. However, patients with altered states of consciousness suffer from atypical clinical patterns which can complicate their treatment and daily care. The brain lesions are indeed widespread and affect the nervous system at various levels (cortical, infracortical and spinal). In addition, patients are often bedridden and lack voluntary movements, which exacerbates the spastic symptoms [7]. However, an optimized and individual management of muscular hypertonicity is therefore crucial since it is known that the presence of spasticity is correlated with pain [3,8,9] and that it could also prevent the patient from showing some signs of consciousness which may increase the rate of diagnostic errors [10].

Aside from palliative treatments, DOC patients face a critical lack of therapeutic options. The only drug that was tested in a randomized controlled trial and showed beneficial effects on the level of consciousness in subacute traumatic patients is amantadine [11]. Several neuromodulation methods are also under investigation [12,13]. Regarding spasticity, there are no guidelines to manage it as is the case for stroke patients or patients with spinal cord injury [14,15]. Given this gap, we here aim to review the spectrum of therapeutic options to manage spasticity for patients with altered state of consciousness following a severe brain injury.

## 2. Materials and Methods

A literature review was conducted on 7 September 2017 using PubMed as an online database. The following PubMed Advanced Search Builder was used: (“disorders of consciousness” (All Fields) OR “coma” (All Fields) OR “vegetative state” (All Fields) OR “minimally conscious state” (All Fields)) AND (“spasticity” (All Fields) OR “motor disorder” (All Fields) OR “hypertonicity” (All Fields)). All English language studies that evaluated the effect of a treatment on the spasticity of DOC patients and/or its occurrence were included. Inclusion criteria were: (1) patients in coma, UWS or MCS; (2) spasticity assessed as a primary or secondary outcome and objectively reported (Modified Ashworth Scale (MAS), Modified Tardieu Scale (MTS), or other related clinical scales) from traumatic or non-traumatic etiologies. Given the scarcity of this topic, no time cut-off was considered for publications and all types of articles were considered (i.e., clinical trial, observational or open-label studies, case-reports, and reviews).

## 3. Results

PubMed database returned 157 articles wherein 139 were excluded for at least one of the following reasons: spasticity not assessed, not on DOC patients, not on spasticity in DOC patients, and not written in English (see Figure 1). Eighteen were kept and divided into three main sections: (1) occurrence of spasticity in DOC patients (3 studies); (2) treatment targeting spasticity in DOC patients (6 studies) and (3) treatment with spasticity as secondary outcome (9 studies).

### 3.1. Occurrence of Spasticity in DOC Patients

We identified 3 prospective cross-sectional studies reporting the incidence of spasticity among DOC patients. Thibaut et al. [3] identified 58 out of 65 (89%) chronic patients with a MAS score ≥1 for at least one limb). Interestingly, a positive correlation was found between the MAS scores and the Nociception Coma Scale-Revised scores assessing pain in DOC patients [16]. Earlier, Nakase-Richardson et al. [17] described a proportion of 70% of spastic patients in a sample of 122 DOC patients (active duty military personnel and veterans) during the neurorehabilitation phase. The prevalence of spasticity was the highest among all the assessed medical complications (e.g., dysautonomia, seizure, shunt placement or heterotopic ossification). Ganesh et al. [18] also investigated the comorbid conditions during rehabilitation to better predict the outcome at 1 year in 157 patients. During their period with altered state of consciousness, 59% suffered from spasticity, which was again the highest comorbidity observed in this sample.

### 3.2. Treatments Targetting Spasticity in DOC Patients

We found 6 articles that focused on the treatment of spasticity: 2 clinical trials, 2 case reports and 2 reviews.

The first clinical trial evaluated the effects of soft splints worn in the hands for 30 min against a 30 min physiotherapy session and a control condition in 17 DOC patients [19]. Both the application of soft splints and the passive stretching significantly (however transiently) reduced the spasticity as measured by the MAS. The soft splint was the only intervention that improved the hand opening. The splint was well tolerated by all the patients.

The second clinical trial investigated the effects of acupuncture on the excitability of the spinal motor neurones as measured by event-related electromyography (F wave, M wave and F/M ratio) in 11 chronic DOC patients due to a traumatic brain injury [20]. A significant decrease in the F/M ratio was found 10 min after needle insertion (*p* < 0.001) as well as 10 min after needle removal (*p* < 0.001) after the acupuncture session, while no significant change was observed after the control session. The authors hypothesize that the decrease in the overexcitability of the spinal motor neurons of the abductor pollicis brevis muscle will induce a decrease in spastic muscle hypertonia; however, no objective measurement of spasticity was conducted in their study.

The first case report presented the case of a 40-year-old male patient in subacute UWS and presenting severe spasticity of the 4 limbs [21]. He received daily intrathecal injections of baclofen (50 µg for the two first weeks then 100 µg) through an epidural catheter for 5 weeks. Spasticity decreased significantly along with a progressive recovery of consciousness as claimed by the authors however no objective assessment was reported. The patient was discharged atfer 2.5 months of hospital stay and could speak coherently and walk with support.

The second case report included 4 acute patients (in an intensive care unit) who failed to respond to conventional treatments targetting spasticity and autonomic disorders (i.e., sedation, β-blockers and oral baclofen) [22]. They also underwent intrathecal injections of baclofen which was then administred continuously (25 µg/mL) for an initial dose of 400 µg/day that was adjusted twice a day. After an adjustment period of 48h, the MAS score was consistently reduced in all the patients (minus 1 point on average for the upper limbs and minus 2.5 points for the lower limbs).

Bentley Leong published two reviews matching our criteria: the first one reviewed the neuropathology of DOC in children along with the pathophysiology of spasticity and related treatment options [23].The second one focused on passive muscle stretch as treatment in DOC children [24]. In the first review, treatment options in the overall management of children in UWS or MCS were discussed. The author classified them into three types of interventions: physical, pharmacological and surgical. However, only neurosurgical procedures have been investigated for DOC children. The first type of intervention (i.e., physical treatments) includes passive range of motion (ROM), prolonged muscle stretching, application of splints, orthoses or serial casts, bed positioning, wheelchair seating, tilt-tables and standing frames. The second type (i.e., pharmacological treatments) is divided into drugs having a local or a systemic action. The local compounds are phenol and botulinum toxin injections. While the phenol acts at the neuromuscular junction by chemical neurolysis, botulinum toxin blocks the neuromuscular conduction by preventing the release of acetycholine. Both can be used in combination; however, the author did not report any study performed in children with DOC assessing the effects of such combination. The pharmalogical systemic treatments are oral baclofen, dandtrolene sodium and diazepam. However, the authors stressed that these systemic medications induce sedation and their clinical efficacy has not been clearly established in DOC patients yet. Finally, the third type of intervention (i.e., neurosurgical procedures) includes intrathecal baclofen (ITB) and selective posterior rhizotomy (SPR). Baclofen administered by an intrathecal pump causes less severe side effects since higher concentrations in the cerebrospinal fluid can be reached using lower doses. The author also reported several studies on ITB efficacy in severely disabled subjects. In this review, Leong included a study performed on 18 patients among whom 12 were in UWS. A decrease in the MAS scores was found in every patient selected for ITB therapy (mean decrease: 2.17 points) [25]. We did not include this study based on our review process since the abstract only presented the studied population as patients with brain injury which was too broad for our inclusion criteria. The author suggested a multimodal approach of these three kinds of treatments (i.e., physical, pharmacological and surgical) as it has been used for adults with severe brain injury [26]. The second review assessed the efficacy of passive muscle stretching in the treatment of children in UWS or MCS. This intervention is widely used by physiotherapists in order to reduce the spasticity and increase ROM. After analysing 17 studies, the author came to the conclusion that there is little evidence on the efficacy of passive range of motion and prolonged muscle stretching.

### 3.3. Treatments with Spasticity as a Secondary Outcome in DOC Patients

In this category, we found 9 studies: 2 clinical trials, 1 open-label study, 4 case reports and 2 reviews.

The first trial investigated the effects of cortical activation by thalamic stimulation (CATS) on the level of consciousness in 3 patients (2 in UWS and one in MCS) [27]. They underwent implantation of bilateral thalamic electrodes and were stimulated for 18–48 months. In addition to the improvement in the Coma Recovery Scale-Revised (CRS-R), the severity of limb spasticity was reduced. However, the authors used the Unified Myoclonus Rating Scale only as an objective assessment. The beneficial effects of this stimulation on spasticity were only subjectively assessed by the team and the families. The second randomized controlled trialevaluated the effects of tilt table therapy (conventional or with an integrated stepping device) on the level of consciousness in 50 subacute DOC patients (UWS or MCS with a mean time since injury of 8 weeks and with both traumatic and non-traumatic etiologies) [28]. A significant improvement (*p* < 0.05) in the CRS-R scores was found for both treatments (i.e., tilt table with and without the integrated stepping device) but was significantly higher for the conventional tilt table (*p* < 0.05). Spasticity was also assessed by the MAS after three weeks of intervention. For both interventions, a worsening was found in 18–22% of the tested muscles, an improvement in 8–9% and no change in 70–74% at 3-week follow up as compared to baseline.

An open-label study investigated the effects of ITB pumps on consciousness, performance status and spasticity in 8 patients with chronic DOC [29]. After ITB therapy (mean period: 38 months), all the patients showed a decrease in their MAS scores with a mean decrease of 3.5 to 1 [29].The first case report described 2 young patients (21 and 26 years) in MCS who developed severe spasticity (MAS = 4) following a traumatic brain injury (TBI) [30]. After receiving ITB therapy for a few months they both emerged from the MCS and showed a drastic reduction of spasticity (MAS score diminished from at least 2 points). The second one described two older patients (43 and 67 years) with subarachnoid hemorrhage who were candidates for ITB therapy as well [31]. The first patient showed a decrease of 1.5 points on average on the MAS scores after 8 months of treatment in addition to the clinical recovery (the patient was able to eat, brush his teeth, stand and speak). The second patient showed a decrease of 1.25 points on average (MAS scores) after 6 months and emerged from the MCS as soon as 1 month after the implantation of the ITB. The third study presented 2 young spastic patients (8 and 18 years) in UWS who also underwent ITB therapy and showed improvement in MAS scores but no further details were provided since the focus of this paper was cognitive recovery rather than spasticity *per se* [32]. The last case report presented 5 chronic patients in UWS who received ITB therapy [33]. After 5 months, the average decrease in MAS score was 1 point. Besides, all the CRS-R scores increased significantly in the first weeks and 2 patients emerged from MCS.

The first review was a perspective on early rehabilitation after severe brain injury [34]. The authors stressed the strong agreement existing between the clinicians of 18 French hospitals in favor of physical therapy (passive mobilization and Bobath concepts [35]) and oral drugs (baclofen, dantrolene sodium and diazepam). Tizanidine also seemed to be increasingly used since it could be better tolerated than baclofen, with similar effects [34]. Indeed, even though ITB has been shown to be effective in spasticity following TBI, according to the authors, potential side effects include vigilance alteration and risk of seizures. They also underlined the benefits of botulinum toxin injections although only a few French clinicians seemed to use it in the early stage. However, the authors did not specifically report studies on the treatment of spasticity for DOC patients. The second review focused on the effect of ITB on spasticity, pain and consciousness in this population and in Locked-in Syndrome [36]. The authors extensively described the improvements in consciousness following ITB therapy without mentioning the improvements observed regarding spasticity.

All the studies reviewed are presented in Table 1.

Regarding interventions decreasing spasticity, six types of treatment were identified through this review (i.e., CATS, ITB, passive muscle stretching, acupuncture, soft splints and tilt table therapy). The anatomical targets can be subdivided into three parts: the diencephalon, the spinal cord and the limbs. Figure 2 presents the location of these treatments as well as other conventional pharmacological treatments.

## 4. Discussion

The aim of this review was to identify and analyze the literature referring to the prevalence and the treatment of spasticity in DOC patients. While it is often stated that spasticity is one of the commonest complications following a severe brain injury, little is known regarding the precise number of DOC patients facing this issue. The small number of articles found in this systematic review highlights how the presence of spasticity along with an altered state of consciousness is poorly studied. Two studies were conducted during the rehabilitation phase (first year after the injury) and report a percentage of spastic DOC patients ranging from 59 to 70% [17,18]. A third study was conducted with chronic patients (mean time since insult: 39 ± 37 months) and reports a higher percentage (89%) with a major proportion a patients presenting severe spasticity (>60%) [3].These results suggest that spasticity not only occurs later at a chronic stage but can also worsen in patients who develop it at an early stage. This study also considered the presence of pain and showed a significant correlation between these two factors, meaning that spastic DOC patients are more likely to be in pain; and since they are not able to express it appropriately, specific attention should be taken regarding nursing and the use of painkillers. Spasticity may sometimes help patients who underwent a stroke or a TBI to stand or to grab by supplying other weak muscles. However, DOC patients only face the negative impacts of spasticity (retractions, loss in ROM, pain) due to their sustained immobilization and the lack of voluntary command.

Regarding therapeutic options for spasticity, several treatments were reviewed by Leong (physical, pharmacological and surgical) for spastic children [23]. None of the reviewed treatments seemed to present great benefits in the management of spasticity, with an exception for ITB therapy [21,22,29,30,31,32,33]. This could be partly explained by the paucity of randomized clinical trials assessing the effects of physical therapy. Additionally, the results were extrapolated from adult populations with stroke, TBI and children with cerebral palsy to DOC patients. In fact, in this review, none of the reviewed studies were specifically conducted in DOC patients. The lack of clinical trials reported by this author in 2002 still continues to be an issue. Indeed, we only found two clinical trials that were published more recently: the first one on the effects of soft splints [19] while the second one tested acupuncture [20]. Soft-splinting seems to be an efficient option to reduce hand spasticity and to improve hand opening. The effects were similar to physiotherapy treatment using passive stretching, while soft splints do not require the presence of a physiotherapist and can be applied several hours per day. Soft splints could thus be a great adjuvant to physiotherapy to counter spasticity. However, clinical trials investigating possible long-term benefits are still lacking. The second study assessed the effects of acupuncture, which is not a common option in Western Europe and the United States but is frequently used in Asia. Here the authors report significant effects following acupuncture with objective/quantitative electromyography measurements. The reduction in F/M ratio was correlated with a decrease in spinal motor neuron excitability. Given the pathophysiology of spasticity, the assumption that this decrease would enable the clinical spasticity (muscle tone) to diminish is valuable, however this still needs to be tested by comparing the F/M ratio with the MAS or another clinical scale.

Regarding the pharmacological approach, two case reports investigated the effects of ITB therapy on spasticity in 5 acute spastic DOC patients (4 in coma and 1 UWS). The administered dose ranged from 100 to 400 µg a day and led to a decrease in spasticity for all the patients. ITB therapy therefore seems to be a valuable therapeutic option even though there is a critical lack of evidence due to nonexistent clinical trials. As stressed by Pistoia et al. [36], this lack is due to practical and economic difficulties in setting up multicenter trials since ITB pumps are extremely expensive as compared to the oral treatment, in addition to the higher risks and adverse effects related to the surgery.

Since spasticity is a complication in DOC patients but is not the main “medical priority”, many studies are focusing on the recovery of consciousness by trying treatments targeting higher cognitive functions, with spasticity as a secondary outcome. ITB therapy, for instance seems to be useful not only to treat spasticity as mentioned above, but it may also have an impact on recovery of consciousness. Indeed, we found more trials (half of the presented studies) referring to dramatic improvements in recovery of consciousness following ITB than trials focusing on muscle tone reduction using ITB. In total, 24 patients were treated (17 UWS and 7 MCS) with ITB and showed not only a reduction in spasticity but also notable cognitive improvements in 5 patients and even consciousness recovery in 9 other patients. The placement of the pump requires an invasive procedure and many follow ups which does not promote its widespread usage. However, the process to check if the patient will respond to this treatment is straightforward and only requires intrathecal injections of baclofen, as described by Shrestha et al. [21]. These injections should therefore be considered in patients presenting severe and global spasticity associated with an altered state of consciousness, even at an early stage [22]. Other valuable options, not targeting spasticity specifically, were identified. For instance, the CATS trial highlighted not only consciousness improvements in 3 patients but the authors also reported an effect on spasticity severity [27]. In contrast, tilt table therapy did not have any effect on spasticity. This intervention therefore does not seem suited to target spasticity while it still showed some consciousness improvements in a subset of subacute DOC patients.

Without any further evidence, treatments for spasticity have to be adapted individually. Based on this review, we would advise to begin physiotherapy as soon as possible, to consider (soft) splinting of the upper limbs, and especially the hands, and to check if patients are responding to ITB. Later on, depending on the patient’s recovery, surgical interventions might be considered in order to improve their functional status.

## 5. Conclusion

In conclusion, a multidisciplinary approach combining physical, pharmacological and surgical treatments is the key to properly manage spasticity in DOC patients. It is a population vulnerable to complications related to spasticity and these patients need a particular attention in their daily therapeutic management. The alteration of their motor function may indeed not only interfere with rehabilitation and nursing but also prevent them from showing signs of consciousness. It is therefore of paramount importance to tackle this motor disorder; the multimodal approach appears well suited for maximizing functional improvements.

## Figures and Tables

**Figure 1 brainsci-07-00162-f001:**
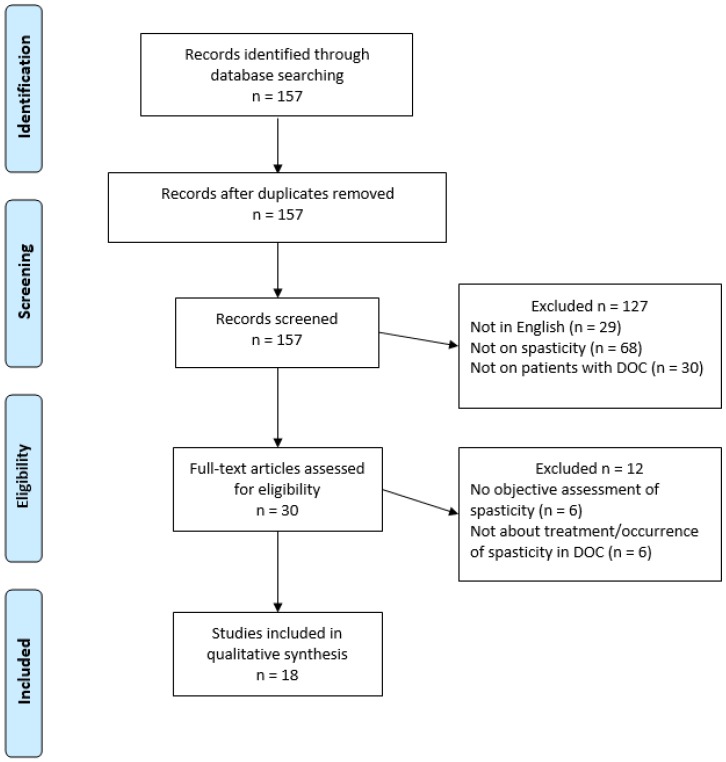
*PRISMA 2009 Flow Diagram*.

**Figure 2 brainsci-07-00162-f002:**
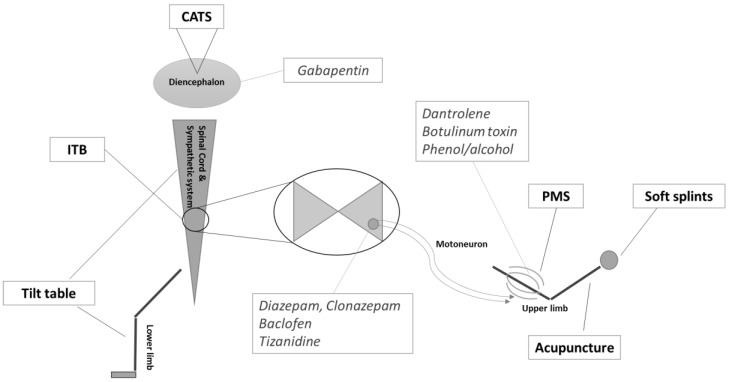
Schematic overview of the reviewed treatments and their anatomical application. CATS = cortical activation by thalamic stimulation; ITB = intrathecal baclofen; PMS = passive muscle stretch. Conventional pharmacological treatments previously described by Thibaut et al. [37] and their anatomical targets are presented as well (in italics).

**Table 1 brainsci-07-00162-t001:** Summary of the reviewed studies. DOC = Disorder of Consciousness; MAS = Modified Ashworth Scale; ITB = Intrathecal Baclofen.

Authors	Study Type	*n*	Intervention	Results
**Occurrence of Spasticity in DOC Patients**				
Thibaut et al., 2014 [3]	Prospective	65	/	89% showed spasticity (MAS ≥ 1)
Nakase-Richardson, 2013 [17]	Prospective	122	/	70% showed spasticity (requiring oral medications, injections or surgical procedure)
Ganesh et al., 2013 [18]	Prospective	68	/	57% showed spasticity
**Treatments targeting spasticity in DOC patients**				
Thibaut et al., 2015 [19]	Randomized controlled trial	17	Soft splints applied to the hand for 30 min	Significant decrease (*p* = 0.014) in the MAS scores (3.5 to 2.5) of the finger flexor muscles
Mastumoto-Miyazaki et al., 2016 [20]	Randomized controlled trial	11	Japanese style acupuncture on 6 points for 10 min	Significant decrease in F/M ratio (*p* < 0.001) of the abductor pollicis brevis
Shrestha et al., 2011 [21]	Case report	1	ITB injections of 50 µg daily for 2 weeks then 100 µg daily for 3 weeks through epidural catheter	Significant decrease of spasticity (subjective report). Patients discharged 2.5 months later, could walk with support
Francois et al., 2001 [22]	Case report	4	ITB injections of 25 µg/mL continuously through intrathecal catheter and then pump	Decrease in MAS score (4.5 to 3.5 for upper limbs and 4.5 to 2 for lower limbs, on average).
Leong, 2002 [23]	Review	/	/	/
Leong, 2002 [24]	Systematic review	17 studies	Passive muscle stretch	Limited evidence
**Treatments with spasticity as secondary outcome**				
Magrassi et al., 2016 [27]	Clinical trial	3	Implantation of bilateral thalamic electrodes	Decrease in spasticity and myoclonus (Unified Myoclonus Rating Scale Section 2 and Section 3: decrease of >25 points)
Krewer et al., 2015 [28]	Randomized controlled trial	50	Tilt table therapy (with our without integrated stepping device) 10 × 1 h	No significant differences (8.5% of all MAS values showed improvement, 19.7% showed worsening and 71.8% showed no difference)
Margetis et al., 2014 [29]	Open-label	8	ITB pump	Decrease in MAS scores from 3.4 to 1, on average
Al-Khodairy et al., 2015 [30]	Case report	2	ITB pump	Decrease in MAS scores (minimum 2 points)
Oyama et al., 2010 [31]	Case report	2	ITB pump (50 µg/day)	Decrease in MAS scores (minimum 1 point)
Taira, 2009 [21]	Case report	2	ITB through lumbar puncture	Effective control of the spasticity
Sara et al., 2009 [36]	Case report	5	ITB pump (100 µg/day)	Decrease in MAS scores (1 point, on average)
Mazaux et al., 2001 [34]	Review	/	/	Good agreement exists among clinicians about prevention of orthopedic complications and treatment for spasticity. However, little consensus exists concerning treatment of non-pyramidal hypertonia and spasms.
Pistoia et al., 2015 [36]	Review	/	/	Although the current indication of ITB is the management of severe spasticity, its potential use in speeding the recovery of consciousness merits further investigation.

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
