# Peer review of "Spasticity Management in Disorders of Consciousness"

_brainsci, 2017, doi:10.3390/brainsci7120162_

Round 1
Reviewer 1 Report
My only comment concerns editing of English language and style; the sense is clear throughout, but it needs to be proofread by someone whose first language is English.
Author Response
Thank you for your advice. We have had the manuscript proofread.
Reviewer 2 Report
The paper deals with a topic of great interest in the relationship between spasticity and DoC in sABI and correctly emphasizes the lack of good quality methodological papers and the inability to reach solid conclusions on the relationship between spasticity reduction and improved consciousness, but also of build evidence-based treatment procedures.
The Introduction is synthetic but effective. In Materials and Methods I suggest specifying to what date the bibliographic search was conducted.
In Results I recommend providing additional data on the papers mentioned to help the reader in making a judgment on their quality. For example:
• Does paper 20 describe the duration of the acupuncture effect?
• In paper 28 what is the interval between acute event and treatment? Is the improvement of consciousness in both groups or not and with significant differences?
In Discussion I would advise you to be more careful in sharing the statement that the tilt table improves your conscience, as it is based on a single low-sample patient study (row265).
Useful spell checker errors, for example errors on line 132, 136,139.
Author Response
We would like to thank the Reviewers for their time in reviewing the manuscript.
- We added the date on which the bibliographic search was conducted to our manuscript
- We added further information on the duration of acupuncture effect for the paper 20
- We added further information on the interval between acute event and treatment as well as on improvement of consciousness to our manuscript
- We rephrased the statement that the tilt table improves your conscience in the discussion
- We checked for spell errors.
Thank you for your useful comments.
Reviewer 3 Report
The authors conducted a systematic review of the occurrence and the therapeutic interventions for patients with disorders of consciousness suffering from spasticity. The work deals with a very spinous problem, which is extensively, clearly, and judiciously faced. In my opinion, the review gives useful information to deal with a very challenging problem that is every day faced by the clinician who deals with patients with DoC.
Author Response
We would like to thank the Reviewers for their time in reviewing the manuscript.
Thank you for your encouraging comments.